# HAS-Flow May Be an Adequate Method for Evaluating Human T-Cell Leukemia Virus Type 1 Infected Cells in Human T-Cell Leukemia Virus Type 1-Positive Rheumatoid Arthritis Patients Receiving Antirheumatic Therapies: A Retrospective Cross-Sectional Observation Study

**DOI:** 10.3390/v15020468

**Published:** 2023-02-08

**Authors:** Kunihiko Umekita, Yuki Hashikura, Akira Takaki, Masatoshi Kimura, Katsumi Kawano, Chihiro Iwao, Shunichi Miyauchi, Takeshi Kawaguchi, Motohiro Matsuda, Yayoi Hashiba, Toshihiko Hidaka

**Affiliations:** 1Division of Respirology, Rheumatology, Infectious Diseases and Neurology, Department of Internal Medicine, University of Miyazaki, Kihara 5200, Kiyotake, Miyazaki 889-1692, Japan; 2Department of Clinical Laboratory, University of Miyazaki Hospital, Kihara 5200, Kiyotake, Miyazaki 889-1692, Japan; 3Institute of Rheumatology, Miyazaki Zenjinkai Hospital, Miyazaki 880-0834, Japan

**Keywords:** adult T-cell leukaemia (ATL), flow cytometry, human T-cell leukaemia virus type 1 (HTLV-1), rheumatoid arthritis

## Abstract

The study aims to assess the usefulness of human T-cell leukemia virus type 1 (HTLV-1)-infected cell analysis using flow cytometry (HAS-Flow) as a monitoring method for adult T-cell leukemia (ATL) development in HTLV-1-positive patients with rheumatoid arthritis (RA) under treatment with antirheumatic therapies. A total of 13 HTLV-1-negative and 57 HTLV-1-positive RA patients participated in this study, which was used to collect clinical and laboratory data, including HAS-Flow and HTLV-1 proviral load (PVL), which were then compared between the two groups. CADM1 expression on CD4+ cells in peripheral blood (PB) was used to identify HTLV-1-infected cells. The population of CADM1+ CD4+ cells was significantly higher in HTLV-1-positive RA patients compared to HTLV-1-negative RA patients. The population of CADM1+ CD4+ cells was correlated with HTLV-1 PVL values. There were no antirheumatic therapies affecting both the expression of CADM1 on CD4+ cells and PVLs. Six HTLV-1-positive RA patients who indicated both high HTLV-1 PVL and a predominant pattern of CADM1+ CD7neg CD4+ cells in HAS-Flow can be classified as high-risk for ATL progression. HAS-Flow could be a useful method for monitoring high-risk HTLV-1-positive RA patients who are at risk of developing ATL during antirheumatic therapies.

## 1. Introduction

Standardization of antirheumatic therapies improves prognosis in patients with rheumatoid arthritis (RA). Several biologics and targeted-synthetic disease-modifying antirheumatic drugs (DMARDs) have transformed rheumatic disease treatment [1]. However, the risk of serious infections and malignancies in patients with RA from the use of biologics and synthetic DMARDs remains unknown [2,3].

Human T-cell leukemia virus type 1 (HTLV-1) is the causative agent of adult T-cell leukemia/lymphoma (ATL) and HTLV-1-associated myelopathy/tropical spastic paraparesis (HAM/TSP). There are approximately 1 million HTLV-1 carriers in Japan, making it endemic, [4] and it is estimated that at least 4000 new HTLV-1 infections occur yearly among adults and adolescents in Japan [5]. Furthermore, the number of HTLV-1 carriers has reportedly increased in urban regions of Japan, such as Chubu, Kanto, and Kinki. [5] Further, the incidence of RA is higher in women than in men; therefore, the prevalence of HTLV-1 infection in RA patients in Japan is expected to be high. Based on the prevalence of HTLV-1 infection and RA, the number of HTLV-1-positive patients with RA in Japan could range from 5000 to 10,000.

The CD4-positive T-cells are the main target of HTLV-1. The incidence rate of ATL was estimated at approximately 5% of HTLV-1 carriers. The risk factors for ATL development have been reported to be advanced age and higher HTLV-1 proviral load (PVL) values (>4.0 copies per 100 peripheral blood mononuclear cells (PBMCs)) in HTLV-1-asymptomatic carriers [6]. An increasing HTLV-1 PVL value, which reflects the oligo- or monoclonal expansion of HTLV-1-infected T-cells, was observed in HTLV-1-asymptomatic carriers before the emergence of ATL [7]. The serum concentration of soluble IL-2 receptor (sIL2-R) reflects the expansion of HTLV-1-infected T-cells [8]. However, quantitative polymerase chain reaction (PCR) measurement of HTLV-1 PVL value is not available in daily clinical practice of HTLV-1 carriers. Standardized quantitative PCR for evaluating HTLV-1 PVL has not yet been commercialized; it is only available for research purposes. Furthermore, using inverse-long PCR methods in clinical laboratory tests to assess clonal expansion of HTLV-1-infected T-lymphocytes is unrealistic. As a result, an uncomplicated and efficacious method for monitoring the dynamics of HTLV-1-infected T-lymphocytes is necessary.

HTLV-1-infected cell analysis system flow cytometry (HAS-Flow) was developed as a novel method for monitoring HTLV-1-infected cells, including ATL cells. The cell adhesion molecule 1 (CADM1) versus CD7 plot accurately reflects disease progression in HTLV-1 infection, and CADM1-positive cells with downregulated CD7 in HTLV-1-asymptomatic carriers have properties similar to those in indolent ATL [9]. For example, CADM1+ CD7-negative (CD7^neg^) cells have been reported to be enriched with clonally expanded cells and abnormal lymphocytes, which were suspected as ATL cells [9]. Several studies that used HAS-Flow analysis suggested an increased risk of developing ATL among participants who had >25% of CD4+ cells that were positive for CADM1 and >50% of CADM1+ cells that were negative for CD7 (CADM1+ CD7^dim^ < CADM1+ CD7^neg^; CADM1+ CD7^neg^-dominant pattern) [9,10,11]. 

However, HAS-Flow has been established as an analysis system for HTLV-1-infected cells using PBMCsobtained from HTLV-1 carriers, including HTLV-1-associated disorders such as HAM/TSP and ATL [9,10,11]. Because no studies have indicated CADM1 expression in CD4 lymphocytes in chronic inflammatory diseases such as RA, it is unclear whether HAS-Flow is an adequate method to evaluate HTLV-1-infected cells and ATL development in HTLV-1-positive RA patients. In addition, it is yet to be elucidated whether antirheumatic therapies including immunosuppressants and biologics effect the expression of CADM1 in CD4 lymphocytes.

Therefore, this study aims to evaluate the usefulness of HAS-Flow in HTLV-1-positive RA patients receiving antirheumatic therapies. In addition, the correlation between HTLV-1 PVL values and the proportion of CADM1+ CD4+ T-lymphocytes in HTLV-1-positive RA patients was examined. The influence of RA disease activity and its antirheumatic therapies on the expression of CADM1 in CD4+ T-lymphocytes were also analyzed. The present study demonstrated that HAS-Flow may be an adequate and useful method for monitoring HTLV-1-infected cells and evaluating the high-risk cell population of ATL progression using HAS-Flow in HTLV-1-positive RA patients.

## 2. Materials and Methods

### 2.1. Study Design and Participants

The HTLV-1 RA Miyazaki registry was conducted from December 2017 at the University of Miyazaki Hospital and Miyazaki Zenjinkai Hospital in the Miyazaki Prefecture, Japan. This registry was started as inheritance research of the HTLV-1 RA Miyazaki Cohort Study [12]. The registry aimed to investigate the effect of HTLV-1 infection on the clinical features of RA patients and to clarify whether immunosuppressive therapies alter the risk factors associated with the development of ATL in HTLV-1-positive RA patients. All participants were diagnosed with RA on the basis of the 1987 diagnostic criteria of the American College of Rheumatology (ACR) and screened for HTLV-1 infection [13,14]. Accordingly, 85 HTLV-1-positive RA patients participated in this registry until April 2022. All RA patients were treated with antirheumatic drugs, such as methotrexate (MTX) and biologic agents, in accordance with RA treatment guidelines [15,16,17]. Written informed consent was obtained from all participants and they were expected to periodically visit the University of Miyazaki Hospital and Miyazaki Zenjinkai Hospitals (formerly named Zenjinkai shimin-no-Mori Hospital) for clinical assessment and sample collection [12].

This registry was used to choose the participants for the current investigation. The following are the study’s inclusion criteria: HTLV-1-positive RA participants who underwent HAS-Flow between April 2019 and December 2021 in this registry. Quantitative PCR was also used to determine HTLV-1 PVL levels. According to these inclusion criteria, 57 of 85 HTLV-1-positive RA patients were enrolled in this study. Furthermore, 13 HTLV-1-negative RA patients had performed HAS-Flow in this registry during the same observation period. All clinical data assessed, such as RA disease activity and antirheumatic regimen, was collected from the patient’s medical records. The study protocol was approved by the research ethics committees of the University of Miyazaki Hospital (approval no. O-0236) and Miyazaki Zenjinkai Hospital, and followed the Ethical Guidelines for Medical and Health Research Involving Human Subjects.

### 2.2. HAS-Flow

For HAS-Flow analysis, whole blood samples in EDTA tubes were obtained periodically from 57 participants with HTLV-1-positive RA and 13 participants with HTLV-1-negative RA. HAS-Flow analysis was performed to detect the expression levels of CADM1 and CD7 in CD4+ T-lymphocytes, according to previously reported methods [11]. An unlabelled CADM1 antibody (clone 3E1) was purchased from MBL (Tokyo, Japan) and subjected to primary amine biotinylation using biotin N-hydroxysuccinimide ester (Sigma Aldrich, St. Louis, MO, USA). All other antibodies were obtained from BioLegend (San Diego, CA, USA). Cells were stained using a combination of biotin-CADM1, allophycocyanin-CD7, and PE-Cy7-CD4. After washing, phycoerythrin (PE)-conjugated streptavidin was applied. A FACS Calibur instrument (BD Immunocytometry Systems, San Jose, CA, USA) was used for all multicolour flow cytometry. Data were analyzed using the FlowJo software (TreeStar, San Carlos, CA, USA).

### 2.3. The Measurement of HTLV-1 PVL

The current study used HTLV-1 PVL levels measured immediately before or after performing HAS-Flow. The HTLV-1 PVL was measured using PBMCs separated by Ficoll-based density gradient centrifugation; DNA was purified from PBMCs; and real-time PCR was performed using the Light Cycler 2.0 thermal cycler (Roche Diagnostics, Mannheim, Germany) to measure the HTLV-1 pX region and the human albumin gene [18]. The PVL levels of HTLV-1 in PBMCs were measured in duplicates.

### 2.4. Clinical Assessment of RA

Clinical information was collected during the observation period while performing HAS-Flow. According to the European League Against Rheumatism (EULAR) guidelines, the Disease Activity Score of 28 joints (DAS28) using the erythrocyte sedimentation rate, Simple Disease Activity Index (SDAI) and Clinical Disease Activity Index (CDAI) were used to evaluate RA disease activity [19]. Physical function was assessed using the Health Assessment Questionnaire-Disability Index (HAQ-DI).

### 2.5. Statistical Analysis

Statistical analyses were performed using EZR (Saitama Medical Centre, Jichi Medical University, Saitama, Japan). EZR is a modified version of R commander (version 2.4-2), which was designed to add statistical functions that are frequently used in biostatistics [20]. The median with interquartile range (IQR) and numbers and percentages were used to express continuous and categorical data, respectively. A group comparison between HTLV-1-negative and HTLV-1-positive RA participants was conducted using the Mann–Whitney U test for age, inflammatory biomarkers value, CDAI, SDAI, DAS28, HAQ–DI, corticosteroid dose, and MTX dose. Categorical variables, including sex, Steinbrocker’s stage and class, rheumatoid factor (RF), anti-citrullinated protein antibody (ACPA) positivity, and usage of corticosteroids and DMARDs such as MTX, tacrolimus and biologic agents, were compared between HTLV-1-negative and HTLV-1-positive RA participants using Fisher’s exact test. A group comparison between HTLV-1-negative and HTLV-1-positive RA participants was conducted using the Mann–Whitney U test for CADM1+ CD4 lymphocytes and the CADM1+ CD7neg CD4 lymphocyte population. The correlation between HTLV-1 PVL, CADM1+ CD4 lymphocytes and CADM1+ CD7neg CD4 lymphocyte population was conducted using Spearman’s test. A *p*-value of <0.05 was considered significant.

## 3. Results

### 3.1. Characteristics of HTLV-1-Negative and HTLV-1-Positive RA Participants

The clinical characteristics of the HTLV-1-negative and HTLV-1-positive RA participants who participated in this investigation are shown in Table 1. Treatment with DMARDs, including MTX and/or biologic agents, was effective in all RA participants, according to the ACR/EULAR guidelines [16]. There were no differences between the two groups in terms of median age, sex ratio, C-reactive protein and erythrocyte sedimentation rate values, or RF/ACPA seroprevalence. According to the EULAR improvement criteria, the CDAI, SDAI and DAS28 values of both groups indicated a low RA disease activity. The proportion of RA participants who received corticosteroids did not change between those who were HTLV-1-negative and those who were HTLV-1-positive. The rate of MTX use, median dosage of MTX and proportion of participants using tacrolimus and biologic agents did not differ between the groups. The number of persons with HTLV-1-associated illnesses such as HAM/TSP was four among HTLV-1-positive RA participants.

### 3.2. The Population of CADM1+ CD4+ Cells in HTLV-1-Negative and HTLV-1-Positive RA Participants

Figure 1 shows representative data of CADM1 versus CD7 plotted by HAS-Flow in HTLV-1-negative RA participants and HTLV-1-positive RA participants. For CD4+ cells, a CADM1 versus CD7 plot was constructed (Figure 1A). According to the previous report, the proportion of CADM1+ CD7dim and CADM1+ CD7neg was named D region (D) and N region (N), respectively [11] (Figure 1B). In HTLV-1-positive RA participants, the population of CADM1+ CD4+ cells (D + N) were higher than in HTLV-1-negative RA participants (15.2% vs. 8.9%, *p* = 0.002) (Figure 2A). Furthermore, the population of CADM1+ CD7neg CD4+ cells (N) in HTLV-1-positive RA participants was higher than that in HTLV-1-negative RA participants (4.3% vs. 2.5%, *p* = 0.006) (Figure 2B). The population of CADM1+ CD4+ cells was lower in HTLV-1-negative RA participants compared to those in HTLV-1-positive RA participants. The effect of antirheumatic therapies on CADM1 expression may be deemed minimal because there were no differences in antirheumatic therapy regimen between the two groups. In addition, there were no associations between the population of CADM1+ CD4+ cells and clinical features of RA, such as disease duration and inflammatory conditions. These findings show that CADM1+ CD4+ cells could be evaluated as HTLV-1-infected cells by HAS-Flow in HTLV-1-positive RA participants.

### 3.3. The Correlation between Population of CADM1+ CD4+ Cells and HTLV-1 PVL in HTLV-1-Positive RA Participants

The median HTLV-1 PVL for the 57 HTLV-1-positive RA participants was 2.09 copies/100 PBMCs (IQR, 5.2). The population of CADM1+ CD4+ cells (D + N) was correlated with the HTLV-1 PVL values in RA participants receiving several antirheumatic therapies (R = 0.44, *p* = 0.0005) (Figure 3A). In addition, the population of CADM1+ CD7neg CD4+ cells (N) was also significantly correlated with the HTLV-1 PVL values (R = 0.6, *p* = 0.0000008) (Figure 3B). In the present study, four RA participants with HAM/TSP were enrolled (Figure 3, open red circles). One of four RA participants with HAM/TSP had a high HTLV-1 PVL of more than 4.0 copies/100PBMCs as well as an increase in the CADM1+ CD7neg CD4+ cell population.

### 3.4. Assessment of High-Risk RA Participants for ATL Development Based on HAS-Flow Analysis and HTLV-1 PVL Values

Table 2 shows the HAS-Flow analysis results of 57 HTLV-1-positive RA participants. Out of the 57 participants, 13 participants (22.8%) had more than 25% of CD4+ cells positive for CADM1. These 13 participants were divided into two groups based on the population of CADM1+ CD7neg cells, with more than 50% among CADM1+ cells (Table 2, a and b). In our analysis, 7 of the 13 participants had an increased number of CADM1+ CD7neg cells (CADM1+ CD7neg dominant pattern) (Table 2, b). Nineteen of 57 participants had high HTLV-1 PVL of more than 4.0 copies/100PBMCs. A CADM1+ CD7neg dominant pattern of HAS-Flow was observed in six participants with high HTLV-1 PVL of more than 4.0 copies/100PBMCs. HTLV-1-positive RA participants with high HTLV-1 PVL and a CADM1+ CD7neg dominant pattern may be classified as high-risk participants with ATL development potential.

## 4. Discussion

This is the first study to show that HAS-Flow analysis may be used to assess HTLV-1-infected CD4+ cells in HTLV-1-positive RA patients under treatment with antirheumatic therapies. 

Among CD4+ cells the population expressing CADM1 was higher in HTLV-1-positive RA patients compared to those in HTLV-1-negative RA patients. CADM1, which was previously discovered as a tumor suppressor in lung cancer [21], has been revealed to be highly and exclusively expressed in HTLV-1-infected cells in the peripheral blood (PB) regardless of the ATL subtype [22,23]. HTLV-1 clones can be highly purified in the CADM1-positive population of CD4+ cells in PB samples from HTLV-1 asymptomatic carriers as well as ATL patients [9,11]. However, there were no reports to clarify the population of CADM1+ CD4+ cells in the PB of RA patients. This study suggests that the population of CADM1+ CD4+ cells was lower in HTLV-1-negative RA patients compared to HTLV-1-positive RA patients. In addition, the percentage of the CADM1+ CD4+ cell population in PB correlated with the HTLV-1 PVL values in HTLV-1-positive RA patients. These HAS-Flow findings corroborated prior data from HTLV-1 carriers, including HAM/TSP patients [24,25]. The mechanisms of CADM1 expression in HTLV-1-infected CD4+ cells are yet unknown. Several studies have suggested that the HTLV-1-related protein Tax is responsible for the expression of CADM1 in HTLV-1-infected cells [24,25]. The Tax-NF-kB complex binds to the CADM1 promotor region [26]. As a result, CADM1 expression in CD4+ cells may be influenced by HTLV-1 Tax activity in both HTLV-1 carriers and HTLV-1-positive RA patients. In addition, our results suggest that antirheumatic therapies may not affect the expression of CADM1 on CD4+ cells in HTLV-1-positive RA patients. Therefore, HAS-Flow analysis may be useful to monitor HTLV-1-infected cells in RA patients receiving antirheumatic therapies.

The independent risk factors for the progression of ATL have been indicated as follows: a higher HTLV-1 PVL, advanced age, family history of ATL, and the first opportunity for HTLV-1 testing during treatment for other diseases [6]. In addition, an increase of more than 25% of CADM1+ CD4+ cells may be one of the risk factors of ATL development in patients with HTLV-1 carriers, including HAM/TSP patients [10,24]. As a result, HTLV-1-positive RA patients with a high PVL value and a significant population of CADM1+ CD4+ cells could be classified as having a high risk of developing ATL. In this study, 19 patients out of 57 HTLV-1-positive RA patients had both a high PVL value and a large population of CADM1+ CD4+ cells, more than 4.0 copies/100PBMCs, and more than 25%, respectively. CD7 downregulation has been reported as a predictive biomarker for progression to ATL [10,24]. The CADM1+ CD7neg dominant pattern of HAS-Flow may be an important finding as predictive ATL progression. In this study, six HTLV-1-positive RA patients showed high HTLV-1 PVL and a CADM1+ CD7neg dominant pattern. These six patients may need close follow-up as high-risk patients with ATL development during antirheumatic therapies. As a result, evaluating both PVL and HAS-Flow, not only for HTLV-1 carriers but also for HTLV-1-positive RA patients, may be a useful method to estimate the risk of ATL development.

A recent study developed and demonstrated the use of oligoclonality index (OCI)-Flow, which is another flow cytometry method to analyze HTLV-1-infected cells, based on the T-cell receptor (TCR) Vβ subunit diversity on T-cells infected with HTLV-1 (CD3+, CCR4+, and CD26− T cells) [27]. OCI-Flow appears to be more specific for identifying patients with ATL-like clones, compared with methods that evaluate CD7 downregulation to predict the development of ATL. Because CD7 downregulation in CD4+ cells increases with increasing PVLs, identification of an ATL-like clone based only on the levels of CD7 expression, such as HAS-Flow, may be limited. Compared with HAS-Flow, OCI-Flow may provide more information on the oligoclonality of HTLV-1-infected cells, however, the versatility and convenience of tests needs to be considered during assessment of the risk of developing ATL; therefore, stratification of these patients using a simple method, such as HAS-Flow, may be considered useful. In the future, a more sensitive and specific prediction algorithm to assess the risk of developing ATL will be developed by combining flow cytometry, such as OCI-Flow and HAS-Flow, next generation sequencing to evaluate gene mutations [10], and a method to track and analyze clonality by detecting transgene integration sites [28].

However, there are several limitations to the present study. Firstly, the sample size was small. Secondly, because this study has been performed as a cross-sectional observational study, there was no time-sequential evaluation of HAS-Flow in this study. Therefore, the impact of antirheumatic therapies on the population of CADM1+ CD4+ cells in patients with HTLV-1-positive RA is yet to be elucidated. It is necessary to investigate whether CADM1+ CD7neg dominant pattern is altered during antirheumatic therapies based on long-term observation. Thirdly, we used HAS-Flow and HTLV-1 PVL to assess the high-risk for ATL development in HTLV-1-positive RA patients; six patients were identified as high-risk in this study. However, whether these high-risk RA patients developed ATL is unknown. As a result, we intend to conduct a large-scale study in the future with prolonged follow-up periods to investigate the utility of HAS-Flow as a monitoring and/or predictive approach for ATL development in HTLV-1-positive RA patients.

## 5. Conclusions

In conclusion, the analysis of CADM1+ CD4+ cells by HAS-Flow may be a useful method to observe HTLV-1-infected cells in HTLV-1-positive RA patients. Antirheumatic therapies, including immunosuppressants and biologics, may have less effect on the expression of CADM1. Because CD7neg HTLV-1-infected cells have a significant potential of ATL progression, examining the CADM1+ CD7neg predominant pattern in HTLV-1-positive RA patients may be a predictive finding of ATL development. However, whether antirheumatic medications affect the progression of ATL in HTLV-1-positive RA patients is unknown. Future research into the time-sequential HAS-Flow study could provide insights into the role impact of antirheumatic drugs in ATL progression.

## Figures and Tables

**Figure 1 viruses-15-00468-f001:**
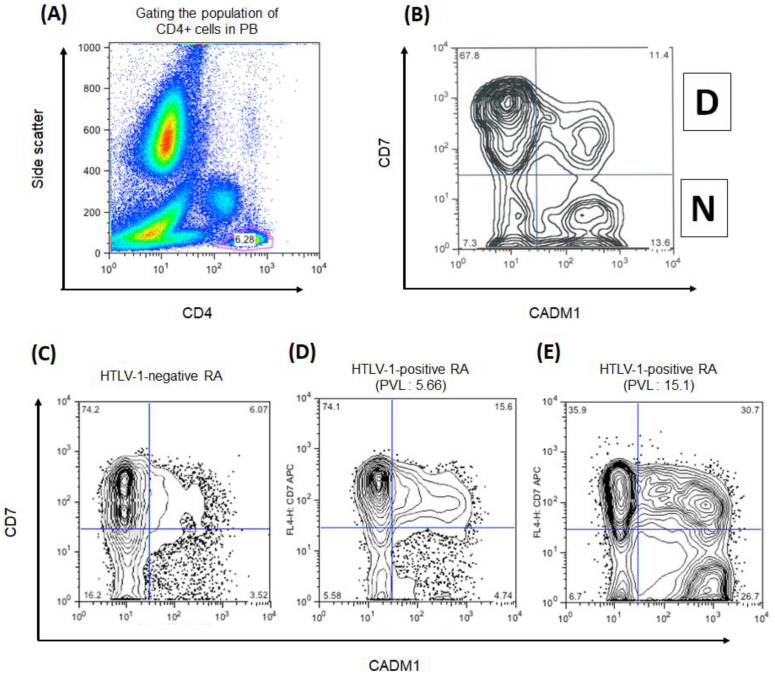
The representative data of CADM1 versus CD7 plot in CD4+ cells in participants with rheumatoid arthritis (RA). (**A**) The gating of the CD4+ cell population in peripheral blood (PB). (**B**) CADM1 versus CD7 plot of CD4+ T-cell population. Upper-right and lower-right regions were named D and N, following previous study. The CD4+ T-cells in D and N region are human T-cell leukemia virus type 1 (HTLV-1)-infected cells. (**C**–**E**) The CADM1 versus CD7 plot in CD4+ cells in HTLV-1-negative RA patient, HTLV-1-positive RA participants with different proviral loads (PVLs). The HTLV-1 PVL were 5.66 and 15.1 copies/100 peripheral blood mononuclear cells, respectively.

**Figure 2 viruses-15-00468-f002:**
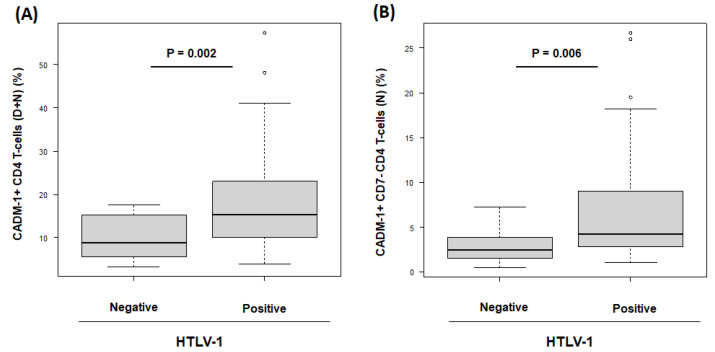
The comparison of CADM1+ CD4+ T-cell populations between human T-cell leukemia virus type 1 HTLV-1-negative and -positive participants with rheumatoid arthritis (RA). (**A**) The box plot of the population of CADM1+ CD4+ T-cells, which present in D + N region, in HTLV-1-negative and -positive RA participants. (**B**) The box plot of the population of CADM1+ CD7neg CD4+ T-cells, which present in N region, in HTLV-1-negative and -positive RA participants.

**Figure 3 viruses-15-00468-f003:**
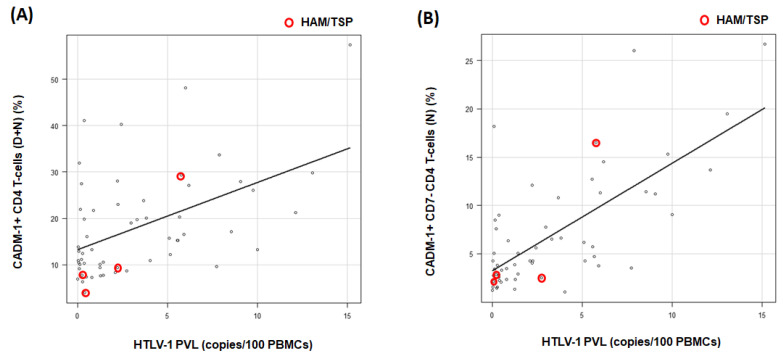
Scatter plot of the human T-cell leukemia virus type 1 (HTLV-1) proviral load (PVL) versus the percentage of CADM1-positive cells from the CADM1 versus CD7 plot. (**A**), X- and *Y*-axis indicates HTLV-1 PVL and the population of CADM1+ CD4+ T-cells (D + N region), respectively (R = 0.44). (**B**), X- and *Y*-axis indicates HTLV-1 PVL and the population of CADM1+ CD7 neg CD4+ T-cells (N region), respectively (R = 0.6). The plot with red circles indicated RA participants with HTLV-1-associated myelopathy/tropical spastic paraparesis (HAM/TSP). PBMCs: peripheral blood mononuclear cells.

**Table 1 viruses-15-00468-t001:** Characteristics of HTLV-1-negative and -positive participants with RA.

	HTLV-1 Negative	HTLV-1 Positive	
	(*n* = 13)	(*n* = 57)	*p*-Value
Age, year (IQR)	67 (19)	70 (12.8)	0.57
Female, no. (%)	10 (76.9)	45 (77.6)	0.99
RA duration, year (IQR) ^a^	13 (19)	11 (15)	0.93
Positive for RF, no. (%) ^b^	61.5	68.1	0.74
Positive for ACPA, no. (%) ^c^	66.7	66.7	0.99
Steinbrocker’s classification, (%)			
Stage Ⅰ/Ⅱ/Ⅲ/Ⅳ	3/4/3/3	5/27/6/18	0.24
Class 1/2/3/4	6/6/1/0	31/10//9/6	0.16
CRP (mg/dl) (IQR)	0.09 (0.2)	0.11 (0.3)	0.57
ESR (mm/60 min) (IQR)	18 (30)	14 (19)	0.68
CDAI (IQR) ^d^	5.00 (5)	2.9 (7.5)	0.98
SDAI (IQR) ^d^	5.56 (5)	3.25 (8.5)	0.99
DAS28 (IQR) ^d^	2.74 (2)	2.27 (1.4)	0.88
HAQ-DI (IQR) ^d^	0 (0)	0.19 (1.6)	0.39
Corticosteroid in use, no. (%)	6 (46)	29 (51)	0.99
Dose of corticosteroid mg/day (IQR) ^e^	4.0 (5)	4.0 (4)	0.94
MTX current user, no. (%)	9 (69)	27 (47)	0.22
Dose of MTX mg/week (IQR)	6.0 (2)	8.0 (4)	0.41
TAC current user, no. (%)	1 (7.7)	13 (23)	0.44
Biologics current user, no. (%)	6 (46)	13 (23)	0.16

Values are expressed as the median with interquartile range (IQR). Percentage (%) is calculated based on the total patient number of each group, unless indicated otherwise. RF: rheumatoid factor; ACPA: anti-citrullinated protein antibody; CRP: C-reactive protein; ESR: erythrocyte sedimentation rate; TJC: tender joint count; SJC: swollen joint count; VAS: visual analog scale; CDAI: clinical disease activity index; SDAI: simplified disease activity index; DAS28: disease activity score in 28 joints; HAQ-DI: health assessment questionnaire-disability index; DMARDs: disease modifying antirheumatic drugs; MTX: methotrexate; TAC: tacrolimus. ^a^ Data available in 13 and 55 patients of the HTLV-1-negative and HTLV-1-positive RA groups, respectively. ^b^ Data available in 13 and 48 patients of the HTLV-1-negative and HTLV-1-positive RA groups, respectively. ^c^ Data available in 12 and 49 patients of the HTLV-1-negative and HTLV-1-positive RA groups, respectively. ^d^ Data available in 12 and 57 patients of the HTLV-1-negative and HTLV-1-positive RA groups, respectively. ^e^ Prednisolone equivalent.

**Table 2 viruses-15-00468-t002:** HAS-Flow results and proportion of HTLV-1-positive RA participants with high proviral load (PVL) values.

	HAS-Flow Analysis	
	No. of samples	percentage, %
No. of HTLV-1-positive RA patients	57	100
CADM1 ≥ 25%	13	22.8
(a) CADM1 + CD7dim > CADM1 + CD7neg	6	10.5
(b) CADM1 + CD7dim < CADM1 + CD7neg	7	12.3
HTLV-1 PVL > 4.0 copies/100PBMCs	19	33.3
with (a)	2	3.5
with (b)	6	10.5

## Data Availability

All data are available under request to the corresponding author.

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
