# Peer review of "HAS-Flow May Be an Adequate Method for Evaluating Human T-Cell Leukemia Virus Type 1 Infected Cells in Human T-Cell Leukemia Virus Type 1-Positive Rheumatoid Arthritis Patients Receiving Antirheumatic Therapies: A Retrospective Cross-Sectional Observation Study"

_viruses, 2023, doi:10.3390/v15020468_

Round 1
Reviewer 1 Report
In the manuscript by Umekita et al, the authors investigate the feasibility of using HAS-Flow to evaluate HTLV-1 infected cells in HTLV-1 positive rheumatoid arthritis (RA) patients receiving antirheumatic therapies. It is a retrospective study in which they determined the ratio of CADM1+ CD4+ CD7dim/neg population in peripheral blood of HTLV-1 negative and positive RA patients. They find, that together with patients with high PVL and a predominant pattern of CADM1+ CD4+ CD7neg determined by HAS-Flow can be classified as high-risk for ATL progression.
As the authors point out, there are some shortcomings in the study. They suggest that RA-treatment doesn’t influence the ratio of CADM1+ CD4+ CD7dim/neg population in HTLV-1 positive vs negative patients, but a kinetics experiment is needed to see whether there is a change over time in the level of CADM1+/CD4+/CD7neg cells. This was appropriately discussed.
I do think it is an important and relevant study, HTLV-1 positive patients need close monitoring and being able to predict whether a carrier is likely to develop ATL is important. Whether and if so how RA treatment influences the chances of ATL progression will be an important follow up study. Recently, a report was published in which a novel method, OCI-Flow was developed in which TCRVbeta subunit diversity in T cells infected with HTLV-1 CD3+ CCR4+ CD26- T cells was shown to be a good predictor of ATL development (OCI-Flow >= 0.77)(more so than the HAS-Flow reporter previously and used in this study) (Wolf, S et al, Blood Cancer Journal 2021). I would expect the authors to at least include this study and discuss their findings in relation to this report in the discussion.
Minor comments:
Line 22: treatment instead of treated
Line 78: Flow is an adequate method
Table 1: Female instead of Femail
Line 265: treatment instead of treated
Line 266-267: this sentence does not make sense to me. Please rephrase.
Lines 288-289: Therefore, HAS-Flow analysis may be useful to monitor…
Author Response
Response to Reviewer #1’s comments
Dear Editor and Reviewers,
Thank you for your careful review and the pertinent and insightful comments. We have revised our manuscript according to the reviewers’ suggestions. All of the revised parts in the document and tables are indicated by red font with underline. We hope that our edits and the responses we provided below satisfactorily addressed all the issues and concerns that you and the reviewers have noted.
Best regards,
Kunihiko Umekita
Reviewer #1:
MS No. viruses-2096178
In the manuscript by Umekita et al, the authors investigate the feasibility of using HAS-Flow to evaluate HTLV-1 infected cells in HTLV-1 positive rheumatoid arthritis (RA) patients receiving antirheumatic therapies. It is a retrospective study in which they determined the ratio of CADM1+ CD4+ CD7dim/neg population in peripheral blood of HTLV-1 negative and positive RA patients. They find, that together with patients with high PVL and a predominant pattern of CADM1+ CD4+ CD7neg determined by HAS-Flow can be classified as high-risk for ATL progression.
Response: Thank you for your suggestion. We agree with your opinion; in the future, we intend to conduct a large-scale study with a longer follow-up period to investigate the utility of HAS-Flow as a monitoring and predictive tool for ATL development in HTLV-1-positive patients with RA.
As the authors point out, there are some shortcomings in the study. They suggest that RA-treatment doesn’t influence the ratio of CADM1+ CD4+ CD7dim/neg population in HTLV-1 positive vs negative patients, but a kinetics experiment is needed to see whether there is a change over time in the level of CADM1+/CD4+/CD7neg cells. This was appropriately discussed.
Response: Thank you for your comment regarding the need for a kinetics experiment to observe changes over time in the level of CADM1+/CD4+/CD7neg cells during RA treatment. We will plan an experiment to clarify this research question in the future.
I do think it is an important and relevant study, HTLV-1 positive patients need close monitoring and being able to predict whether a carrier is likely to develop ATL is important. Whether and if so how RA treatment influences the chances of ATL progression will be an important follow up study. Recently, a report was published in which a novel method, OCI-Flow was developed in which TCRVbeta subunit diversity in T cells infected with HTLV-1 CD3+ CCR4+ CD26- T cells was shown to be a good predictor of ATL development (OCI-Flow >= 0.77)(more so than the HAS-Flow reporter previously and used in this study) (Wolf, S et al, Blood Cancer Journal 2021). I would expect the authors to at least include this study and discuss their findings in relation to this report in the discussion.
Response: Thank you for your suggestion. In the Discussion section of the manuscript (page 9, lines 312–327), we described OCI-Flow, which is another novel flow cytometry method that is based on TCR Vβ subunit diversity and appears to be more specific for identifying patients with ATL-like clones, compared with methods that evaluate CD7 downregulation to predict the development of ATL. Since CD7 downregulation in CD4+ cells increases with increasing PVLs, identification of an ATL-like clone based on only the levels of CD7 expression, such as HAS-Flow, may be limited. Compared with HAS-Flow, OCI-Flow may provide more information on the oligoclonality of HTLV-1-infected cells, but the versatility and convenience of tests need to be considered during assessment of the risk of developing ATL; therefore, stratification of these patients using a simple method, such as HAS-Flow, is considered useful. In the future, a more sensitive and specific prediction algorithm to assess the risk of developing ATL will be developed by combining flow cytometry, such as OCI-Flow and HAS-Flow; next generation sequencing to evaluate gene mutations [10]; and a method to track and analyze clonality by detecting transgene integration sites [28].
During the manuscript revision, we added new references [14] and [28].
Minor comments:
Line 22: treatment instead of treated
Response: We have corrected it.
Line 78: Flow is an adequate method
Response: We have corrected it.
Table 1: Female instead of Femail
Response: We have corrected it.
Line 265: treatment instead of treated
Response: We have corrected it.
Line 266-267: this sentence does not make sense to me. Please rephrase.
Response: We agree with your suggestion and deleted this redundant sentence.
Lines 288-289: Therefore, HAS-Flow analysis may be useful to monitor…
Response: We have corrected it.
Reviewer 2 Report
In their study the authors aimed to assess the utility of flow cytometry as a monitoring tool for adult T-cell leukemia (ATL) development in HTLV-1-positive patients with rheumatoid arthritis (RA) during disease-modifying therapies, through measuring the percentage CADM1+ CD7neg CD4+ cells. According to the data of CADM1 versus CD7 plotted by HAS-Flow, in HTLV-1-negative RA patients the percentages of CADM1+ CD4+ cells and of CADM1+ CD7neg CD4+ cells are higher (median values of 8.5% and 2.5%, respectively) than those previously reported in normal controls (Kobayashi et al, Clin Cancer Res. 2014), so it cannot be assumed that the disease itself and the antirheumatic therapies have no impact on CADM1+ expression and CD7 downregulation. A lot of normal controls should have been used in the current study.
Also, the information concerning the expression patterns of CADM1 that is presented in the Material and Methods section should be included in the Introduction.
Author Response
Response to Reviewer #2’s comments
Dear Editor and Reviewers,
Thank you for your careful review and the pertinent and insightful comments. We have revised our manuscript according to the reviewers’ suggestions. All of the revised parts in the document and tables are indicated by red font with underline. We hope that our edits and the responses we provided below satisfactorily addressed all the issues and concerns that you and the reviewers have noted.
Best regards,
Kunihiko Umekita
Reviewer #2:
In their study the authors aimed to assess the utility of flow cytometry as a monitoring tool for adult T-cell leukemia (ATL) development in HTLV-1-positive patients with rheumatoid arthritis (RA) during disease-modifying therapies, through measuring the percentage CADM1+ CD7neg CD4+ cells. According to the data of CADM1 versus CD7 plotted by HAS-Flow, in HTLV-1-negative RA patients the percentages of CADM1+ CD4+ cells and of CADM1+ CD7neg CD4+ cells are higher (median values of 8.5% and 2.5%, respectively) than those previously reported in normal controls (Kobayashi et al, Clin Cancer Res. 2014), so it cannot be assumed that the disease itself and the antirheumatic therapies have no impact on CADM1+ expression and CD7 downregulation. A lot of normal controls should have been used in the current study.
Response: Thank you for your insightful opinion. The median percentages of CADM1+ CD4+ cells and of CADM1+ CD7neg CD4+ cells in patients with HTLV-1-negative RA in our present study seems to be higher (8.5% and 2.5%, respectively), compared with the value in normal controls in a previous report (Kobayashi et al, Clin Cancer Res. 2014). However, we thought that differences in flow cytometry (FCM) conditions and antibody clones used in research would affect the measurement results of FCM, so we think that it has to be careful when comparing flow cytometry data from different studies. In our present study, because healthy individuals were not enrolled in our present study and in the Miyazaki HTLV-1 RA registry study, we could not carry out your suggestion regarding additional comparison of the percentages of CADM1+ CD4+ cells and of CADM1+ CD7neg CD4+ cells between patients with HTLV-1-negative RA and healthy subjects. The roles of CADM-1 and CD7 expressed on CD4+ cells in the pathogenesis of RA remain unclear. Hence, we believe that future research may be necessary to investigate the effect of immunosuppressive therapy on the expression of these molecules on CD4+ cells in patients with RA.
Also, the information concerning the expression patterns of CADM1 that is presented in the Material and Methods section should be included in the Introduction.
Response: We revised our manuscript according to your suggestion. Information on the expression patterns of CADM1 was presented in the Introduction (page 2, lines 72–78).
Reviewer 3 Report
Umekita et al examined and compared the expression patterns of CADM1 and CD7 in CD4+ cells in HTLV-1 negative and HTLV-1 positive patients with rheumatoid arthritis (RA) to assess the usefulness of HTLV-1-infected cells analysis using flow cytometry (HAS-Flow). While majority of HTLV-1-carriers remain asymptomatic, they have a risk of developing HTLV-1-associated diseases, such as ATL and HAM/TSP, through their lives. However, it remains unclear if the other diseases such as RA and the clinical treatment might affect HTLV-1 reactivation, host immune responses against HTLV-1 and development of HTLV-1-associated diseases. Since the expression patterns of CADM1 and CD7 in CD4+ T cells have been reported to be useful to estimate clonal expansion of ATL cells and to predict the disease progression to ATL, it is important to evaluate the assay and to predict the risk of progression from an asymptomatic to a symptomatic state associated with HTLV-1-infection in RA patients with HTLV-1 infection.
1. In Materials and Methods, there is no clear description about HAS-flow, such as sample (EDTA-blood or isolated PBMC), antibodies and instrument.
2. In this study, four RA subjects with HAM/TSP were included in the cohort of HTLV-1-positive RA subjects. Are these subjects comparable to the other HTLV-1-positive RA patients at the levels of the HTLV-1 PVL and the other immunological markers?
3. Is there any association of the population of CADM1+CD4+ cells and HTLV-1 PVLs with the disease duration of RA?
4. The authors described that there were no antirheumatic therapies affected both population of CADM1+CD4+ cells and PVLs, but it is also described that all the RA subjects enrolled in the study were treated with antirheumatic drugs. How did the authors confirm if antirheumatic therapies did not affect both population of CADM1+CD4+ cells and PVLs in this study? Is there any additional supportive results at pre and post treatment?
5. Minor point: Table 1 (Female).
Author Response
Response to Reviewer’s comments
Dear Editor and Reviewers,
Thank you for your careful review and the pertinent and insightful comments. We have revised our manuscript according to the reviewers’ suggestions. All of the revised parts in the document and tables are indicated by red font with underline. We hope that our edits and the responses we provided below satisfactorily addressed all the issues and concerns that you and the reviewers have noted.
Best regards,
Kunihiko Umekita
Reviewer #3:
Umekita et al examined and compared the expression patterns of CADM1 and CD7 in CD4+ cells in HTLV-1 negative and HTLV-1 positive patients with rheumatoid arthritis (RA) to assess the usefulness of HTLV-1-infected cells analysis using flow cytometry (HAS-Flow). While majority of HTLV-1-carriers remain asymptomatic, they have a risk of developing HTLV-1-associated diseases, such as ATL and HAM/TSP, through their lives. However, it remains unclear if the other diseases such as RA and the clinical treatment might affect HTLV-1 reactivation, host immune responses against HTLV-1 and development of HTLV-1-associated diseases. Since the expression patterns of CADM1 and CD7 in CD4+ T cells have been reported to be useful to estimate clonal expansion of ATL cells and to predict the disease progression to ATL, it is important to evaluate the assay and to predict the risk of progression from an asymptomatic to a symptomatic state associated with HTLV-1-infection in RA patients with HTLV-1 infection.
- In Materials and Methods, there is no clear description about HAS-flow, such as sample (EDTA-blood or isolated PBMC), antibodies and instrument.
Response: Thank you for your suggestions. We described the samples in the Materials and Methods, as follows: “whole blood samples in EDTA tubes were obtained periodically from 57 participants with HTLV-1-positive RA and 13 participants with HTLV-1-negative RA” (page 3, lines 125–127).
In addition, we described HAS-flow in the Materials and Methods, as follows:
“An unlabeled CADM1 antibody (i.e., clone 3E1) was purchased from MBL (Tokyo, Japan) and subjected to primary amine biotinylation using biotin N-hydroxysuccinimide ester (Sigma Aldrich, St. Louis, MO, USA). All other antibodies were obtained from BioLegend (San Diego, CA, USA). Cells were stained using a combination of biotin-CADM1, allophycocyanin-CD7, and PE-Cy7-CD4. After washing, phycoerythrin (PE)-conjugated streptavidin was applied. A FACS Calibur instrument (BD Immunocytometry Systems, San Jose, CA, USA) was used for all multicolor flow cytometry. Data were analyzed using the FlowJo software (TreeStar, San Carlos, CA, USA).” (page 3, lines 129–136).
- In this study, four RA subjects with HAM/TSP were included in the cohort of HTLV-1-positive RA subjects. Are these subjects comparable to the other HTLV-1-positive RA patients at the levels of the HTLV-1 PVL and the other immunological markers?
Response: Since only four patients with HAM were in this registry, we did not the compare clinical findings, immunological characteristics, PVL, etc. between patients with RA with and without HAM. We judged that it would be difficult to make comparisons to find out whether there is equivalence in clinical and immunological characteristics between them. In the future, we would like to register more patients with RA with HAM and compare them with those without HAM, in terms of clinical pathology and immunological characteristics, as you suggested.
- Is there any association of the population of CADM1+CD4+ cells and HTLV-1 PVLs with the disease duration of RA?
Response: Thank you for your suggestion. We re-evaluated and found that the population of CADM1+CD4+ cells and HTLV-1 PVLs had no significant correlations with the duration of RA. Accordingly, we included the following sentence in the Results section: “there were no associations between the population of CADM1+ CD4+ cells and the clinical features of RA, such as disease duration and inflammatory conditions (data not shown)” (page 6, lines 216–218).
- The authors described that there were no antirheumatic therapies affected both population of CADM1+CD4+ cells and PVLs, but it is also described that all the RA subjects enrolled in the study were treated with antirheumatic drugs. How did the authors confirm if antirheumatic therapies did not affect both population of CADM1+CD4+ cells and PVLs in this study? Is there any additional supportive results at pre and post treatment?
Response: Thank you for your comment. We described that antirheumatic therapies may not affect the expression of CADM1 on CD4+ cells in patients with HTLV-1-positive RA (page 8, lines 290–291, indicated with underline). Since this was a cross-sectional observation study, we did not investigate the impact of antirheumatic therapies on the population of CADM1+ CD4+ cells in patients with HTLV-1-positive RA. Therefore, as a limitation, we mentioned that the impact of antirheumatic therapies on the population of CADM1+ CD4+ cells in patients with HTLV-1-positive RA is yet to be elucidated (page 9, lines 330–332). Future investigations on time-sequential HAS-Flow should be conducted to clarify this issue.
- Minor point: Table 1 (Female).
Response: We have revised it.
Round 2
Reviewer 2 Report
The authors gave pertinent explanation concerning the design of their study and made the recommended changes in the manuscript.
A few corrections in the Discussion section are necessary before publishing the article:
- - the phrase “The population of CADM1 in CD4+ cells was higher in HTLV-1-positive RA patients compared to those in HTLV-1-negative RA patients CADM1+ CD4+ T-cells” is meaningless (a suggestion: Among CD4+ cells the population expressing CADM1 was…)
- - in the phrase “The Tax-NF288 kB complex binds to the CADM1 promotor lesion” the word “lesion” should be replaced with “region”
- - remove/reformulate the phrase “It considered that the impact of antirheumatic drugs may be less to the activity of Tax-NF293 kB complex on CADM1 expression”